# Display of WEDM Quality Indicators of Heat-Resistant Alloy Processing in Acoustic Emission Parameters

**DOI:** 10.3390/s23198288

**Published:** 2023-10-07

**Authors:** Sergey N. Grigoriev, Mikhail P. Kozochkin, Vladimir D. Gurin, Alexander P. Malakhinsky, Artur N. Porvatov, Yury A. Melnik

**Affiliations:** Department of High-Efficiency Processing Technologies, Moscow State University of Technology “STANKIN”, Vadkovskiy per. 3A, 127055 Moscow, Russia; s.grigoriev@stankin.ru (S.N.G.); v.gurin@stankin.ru (V.D.G.); a.malakhinsky@lism-stankin.ru (A.P.M.); porvatov_artur@mail.ru (A.N.P.); yu.melnik@stankin.ru (Y.A.M.)

**Keywords:** acoustic emission, wire EDM, spectral analysis, monitoring, diagnostic parameters, white layer, discharge pulses, accelerometers

## Abstract

The widespread nature of heat-resistant alloys is associated with the difficulties in their mechanical machining. It forces the use of the wire electrical discharge machining to be wider. The productivity, roughness, and dimensions of the modified layer of the machined surfaces are indicators of the machining quality. The search for new diagnostic parameters that can expand the information content of the operational monitoring/diagnostics of wire electrical discharge machining and accompany the currently used electrical parameters’ data is an urgent research task. The article presents the studies of the relationship between the parameters of acoustic emission signals accompanying wire electrical discharge machining of heat-resistant alloys, process quality indicators, and characteristics of discharge pulses. The results are presented as mathematical expressions and graphs demonstrating the experimentally obtained dependencies. The research focuses on the formed white layer during wire electrical discharge machining. Pictures of thin cross-sections of the machined surfaces with traces of the modified layer are provided. The issues of crack formation in the modified layer and base materials are considered.

## 1. Introduction

Electrical discharge machining refers to technologies based on exposure to the surface of the workpiece with CEF (concentrated energy flows) [1,2,3,4]. The use of this technology is dictated by a modern approach to the manufacturing of products from materials that are difficult to process by traditional methods [5,6,7]. The main advantage of EDM technologies is that they allow for the processing of any conductive materials from which parts of the aviation and space industry are made. These parts often have a complex shape and are made of heat-resistant alloys that are difficult to process with a blade tool. During EDM, a series of electrical discharges are formed in the gap between the tool electrode and the surface of the workpiece. Each discharge causes the local melting and evaporation of electrode materials, forming a new surface. The obtained properties of the surface texture depend on the electrode materials, the type of dielectric, and the parameters of the applied pulses [8,9,10]. All the conditions that determine the method of introducing thermal energy into the interelectrode gap form the rate of removal of the workpiece material and the quality indicators of the resulting surface of the workpiece and the surface layer. Concentrated energy flows have a significant impact on the properties of the surface: its resistance to abrasion, corrosion resistance, and thermal effects [11,12,13].

EDM technology research is traditionally focused on increasing the performance of the process and improving the quality of the resulting surfaces. We should also not forget about the most critical aspect of research aimed at deepening knowledge about the physical phenomena that accompany the EDM processes. This knowledge makes it possible to develop algorithms for finding the most rational processing modes and contribute to the creation of automated equipment built in sections that operate without the participation of operators. The design of such equipment requires its saturation with the means of obtaining information, with the help of which it is possible to obtain real-time information about the current state of the treatment process. This is necessary for the automatic correction of modes and the elimination of negative situations associated with short circuits [5], with conditions that threaten to break the wire electrode and with a decrease in the quality of the resulting surfaces [14,15,16]. The electrical parameters of discharge pulses generated in the interelectrode gap are traditionally used as a source of information about the progress of the workflow in the equipment for EDM. Based on this information, the pulse utilization coefficient is determined, by which it is possible to judge the state of the interelectrode gap and regulate the electrode feed rate [17,18,19,20]. This approach is used in most modern WEDM machines. However, this approach has its drawbacks, leading to the fact that, in the natural process, the treatment is accompanied by both short circuits and breaks of the wire electrode. The situation worsens when processing new materials with low electrical conductivity, such as cermet [5,10]. To study the features of the WEDM of such materials, and to expand the informative capabilities of monitoring systems, researchers are trying to attract new sources of information about the course of the technological process.

The possibility of using temperature control of the working surfaces of the electrodes and monitoring the hydraulic resistance of the interelectrode gap was investigated in the 1970s [20]. Still, the complexity of the implementation of these methods did not allow for their widespread use in modern WEDM equipment. As additional sources of information on the course of the WEDM process in the literature, one can find works concerning the analysis of acoustic emission signals accompanying the process [21,22,23,24].

The presented article describes the results of experiments with the EDM of a nickel-based heat-resistant alloy NiCr20TiAl (standards EN 10269, DIN 17742). The alloy is used to manufacture bars, disks, rings, turbine blades, and other parts operating at temperatures up to 750 °C. The alloy is used in turbine seals, shut-off valves, heaters, and control devices of various reactors. During the EDM of the alloy, the parameters of the applied pulses were varied; during processing, the voltage, discharge current, and acoustic emission signals were recorded [21,25]. During processing, the average wire electrode feed rate was determined; after processing, the parameters of the resulting surface roughness, the volume of the removed material, and the volume of the white layer formed per unit of time were determined [26,27,28].

Acoustic emission signals (AE signals) are widely used in mechanical engineering to monitor the condition of nodes and predict incipient defects. In the literature, you can find many articles with research results on the possibilities of monitoring the technological processes of traditional treatment by monitoring acoustic signals [29,30,31,32]. It was shown in [2,5] that acoustic monitoring can also be applied to technologies for processing with concentrated energy flows. These are laser processing, electron beam doping, and EDM. At the same time, it was noted that VA monitoring can be carried out in different frequency ranges, which makes it possible to monitor changes in the power density of thermal energy entering the surface of the workpiece in order to evaluate the displacement of the technological process toward the melting of the material or vaporization and plasma formation [33,34,35,36].

The purpose of this work was to study the interrelations of the acoustic emission parameters accompanying EDM with the processing output characteristics, which include the performance and quality of the resulting surface.

As elements of the scientific novelty of the presented material, we can note the identified connections between the parameters of acoustic signals and the quality characteristics of WEDM processing: the rate of formation of white layer volumes, the rate of removal of workpiece material, and the roughness of the resulting surface. The described phenomenon of the formation of cracks in the white layer in the boundary zones between the grains of the processed material can also be attributed to scientific novelty.

## 2. Materials and Methods

Experimental studies were conducted to obtain information about the possibility of acoustic emission signal parameters displaying changes in the WEDM process that affect the processing performance and quality indicators of the resulting surface. The heat-resistant nickel-based alloy NiCr20TiAl was used as the processed material in the experiments; the chemical composition is shown in Table 1.

The blanks made of heat-resistant alloy were plates with dimensions of 80 × 20 × 6 mm; CuZn35 brass wire with a diameter of 0.25 mm was used as a wire electrode. Deionized water was used as the working fluid. Processing was carried out on a WEDM machine CUT 30 P (GF AgieCharmilles SA, Losone, Switzerland). The processing area of the machine and the connection diagram of the equipment for recording AE signals are presented in Figure 1.

Figure 1 shows that the accelerometer is installed on the machine table at an elevation that allows it to remain dry after the workpiece is immersed in the working fluid. The accelerometer was fastened with a magnet. Table 2 shows the characteristics of the accelerometer, and Table 3 shows the main characteristics of the E20-10 analog-to-digital converter, which made it possible to record signals with a frequency of 1 MHz.

Table 2 shows that the linear measurement range of vibration acceleration signals extends up to 15 kHz. But for a comparative assessment of the vibroacoustic activity of EDM processes, higher frequency ranges can also be used, including the resonance zone and higher frequency ranges up to 70 kHz. In addition to AE signals, discharge current signals were recorded on the machine using the LA 100-P current sensor (Table 4).

To build mathematical models that relate the parameters of AE signals, processing modes, and characteristics of the quality of the EDM process, a series of experiments was carried out, consisting of cutting a groove in a heat-resistant alloy plate in different processing modes. When varying the modes, different values of the current pulse time (T_on_) and pulse amplitude (I) were set; other factors were kept constant, including the location of the accelerometer. The time between pulses did not change; in all experiments, it was 200 μs. The voltage was also constant 84 V. To evaluate the performance in each experiment, the distance covered by the wire electrode-tool in 30 s was recorded (slot length), and the average slot width was measured. The product of the width of the slot and its length corresponds to the area of the slot, which, at a constant thickness of the workpiece, is proportional to the extracted volume of material in 30 s. Accordingly, this volume, converted into a unit of time, determines the performance of the process (S). Table 5 shows the modes used in each experiment and indicates the performance value S as a percentage of the maximum value obtained in experiment No. 16. The value of S was used to evaluate the performance of WEDM in each experiment.

During processing, acoustic emission signals were continuously recorded. Figure 2 shows the spectra of acoustic signals for experiments No. 5 and No. 16, where the performance was minimal and maximal.

The spectra in Figure 2 show that, in the entire frequency range, the amplitudes of AE at maximum performance are more significant than those at low performance. It was stated in [2,5] that amplitude changes in different frequency ranges can display other phase transitions when a substance is exposed to a concentrated energy flow (CEF). In this regard, three frequency ranges were selected, and the informative capabilities of them had to be evaluated. The choice of ranges was made based on the analysis of AE signal spectra. Frequency ranges were selected where amplitude changes were most pronounced. This usually happens near spectral maxima, which depend on the natural frequencies of the elastic system of the machine, where the workpiece and the accelerometer are located. In Figure 2, the first local maximum is present around 6 kHz. At higher frequencies, local maxima can be noted around 10 and 16 kHz. This is followed by a series of small-amplitude local maxima, of which it is difficult to single out one. Eventually, three frequency ranges were allocated: 4–8; 9–23; and 23–33 kHz. These ranges were named low-frequency, mid-frequency, and high-frequency ranges. The root-mean-square (RMS) amplitudes for these ranges were named A_1_, A_2,_ and A_3_, respectively. Table 6 for 16 experiments shows the RMS amplitudes for all selected ranges. The amplitude values are percentages of the maximum value in 16 experiments. This is carried out for a more visual representation on the graphs of parameters that have a significant discrepancy in absolute units of measurement.

As shown in [2,5], AE signals accompany not only the processes of the melting and evaporation of materials but also the processes of the propagation of thermal stress waves, phase rearrangement, and crystallization. In this work, an attempt was made to relate the formation processes of a white layer (WL) on the workpiece surface after WEDM with the performance and parameters of the AE signals. To accomplish this, in each experiment, sections were made perpendicular to the position of the wire electrode during processing, from which photographs were taken using an Olympus BX51M optical microscope (Ryf AG, Grenchen, Switzerland). Figure 3 shows examples of such pictures, where the WL thickness is shown for individual sections. Each section was pre-polished and etched with a 5% solution of nitric acid in alcohol.

Figure 3 shows that the WL surface is inhomogeneous. It is difficult to describe it by the layer thickness. The photographs show pores and loose formations on the WL surface. However, it is the WL thickness that is more often used as its quantitative characteristic [6,8,12,19,20]. To understand the relation between WEDM performance and WL formation intensity and AE signals, the WL volume formation rate (V_w_) was estimated. Since the performance of WEDM is evaluated by the volumes of material removed per unit of time, it is more correct to also assess the intensity of WL formation by the volumes formed per unit of time. This will be the WL volume formation rate (V_w_).

To evaluate V_w_, the WL area was determined using the special software JMicroVision (Version V 1.3.4). With the help of this software, the perimeter of the white layer was outlined in the photo, and the area of the figure inside the perimeter in pixels was calculated. This area is proportional to the volume WL, since the thickness of the workpiece was constant. Figure 4 shows an example of a photograph of a WL with a perimeter circled.

Since the data on the length of the groove cut in 30 s is known, the size of the surface area in the photographs of all experiments is constant and amounts to 197.4 μm. From these data, it is possible to calculate a value proportional to the volume of the white layer formed per unit of time. Here, we have to proceed from the assumption that the area WL remains constant throughout the thickness of the workpiece being processed. Table 7 presents the initial data for all 16 experiments and the values of V_w_ as a percentage of the maximum value.

Based on the records of current signals, a selective assessment of the maximum values of the time and amplitude of the applied current pulses was carried out. The RMS of these parameters for each experiment are summarized in Table 8.

As can be seen in Table 7, the real values do not coincide with the given ones and have a significant spread. In this regard, as a parameter reflecting the energy of the process, the WEDM performance obtained during the workpiece processing was used.

## 3. Results and Discussions

The information in Table 5, Table 6 and Table 7 contains data about the relations between the parameters of AE signals and the parameters of discharge pulses and the quality characteristics of the WEDM process of heat-resistant alloy treatment. To obtain these relations in the form of mathematical expressions and graphs, the experimental planning technique implemented using the Matcard 14.0.0.163 software was used.

### 3.1. The Relations between the WL Formation Rate, Performance, and AE Parameters

The processing of the data of Table 5, Table 6 and Table 7 using the experimental planning methodology allowed us to construct mathematical models and graphs showing the relation of AE parameters with the rate of formation of the white layer (V_w_). Figure 5 shows the dependence of V_w_ (A_1_, A_2_) in general form and the form of projections of individual sections on coordinate planes. In Figure 5a,b, the colors of the rainbow emphasize the curvature of the surface: a shift from blue to red indicates the convexity of the surface. The same principle holds true for the other color graph figures in this article.

As a result of mathematical processing, the analytical dependence V_w_ (A_1_, A_2_) and its graphic images were built. Since the resulting mathematical model made it possible to extrapolate the dependencies to areas remote from the area of the experiments, the areas where the real experimental points were located are marked with yellow polygons in the graphs in Figure 5. These areas can be treated with the greatest confidence. Respectively, the yellow perimeter can be called the confidence perimeter (CP). Outside of the CP, the results of mathematical modeling should be taken into account with caution. The mathematical model is represented by expression (1):V_w_ (A_1_, A_2_) = 4.4418 − 0.5787·A_1_ + 0.3365·A_2_ + 0.0079·A_1_^2^ − 0.0122·A_2_^2^ + 0.0173·A_1_·A_2_(1)

The standard deviation (SD) was 9.1%. An analysis of the graphs in Figure 5 shows that with the growth of A_1_, the rate of increase in the volume of WL steadily increases (Figure 5c). At V_w_ ˂ 20%, the influence of A_2_ is observed, but at higher values of V_w_, the impact of A_2_ decreases. The effect of A_2_ is clearly seen in Figure 5d. This influence is extreme, but the extremum is rather sluggish. In the region of CP, variations of A_2_ at a high WL growth rate change V_w_ within 10%. A slight decrease in V_w_ with increasing A_2_ is especially noticeable at a low V_w_. This can be explained by the increased fraction of the evaporated substance noted in [2,5].

By analogy with Figure 5, the dependence V_w_ (A_1_, A_3_) was built, and the graphical constructions are shown in Figure 6.

The mathematical model V_w_ (A_1_, A_3_) is represented by expression (2):Vw (A_1_, A_3_) = 9.1252 + 0.6300·A_1_ − 1.1097·A_3_ + 0.0084·A_1_^2^ + 0.0117·A_3_^2^ − 0.0064·A_1_·A_3_(2)

The SD model error (2) was 11.6%. The analysis of dependencies in Figure 6 shows that the influence of A_1_ remains dominant, and the impact of A_3_ has changed compared to that of A_2_. An increase in A_3_ in the CP area leads to a decrease in V_w_. This reduction is within 10% at large values of V_w_. For small values of V_w_, this decrease is more noticeable. If, in Figure 5d, the decline in V_w_ began at large values of A_2_, then, in Figure 6d, the reduction in the growth rate of WL begins with small values of A_3_. This suggests that the intensification of the evaporation of the material can contribute to a reduction in the rate of WL formation. The increase in the amplitude of the AE components in the high-frequency range relative to the amplitudes at low frequencies indicates an increase in the power density of the incoming energy. In this case, the proportion of material removed due to evaporation and ionization increases [2,5].

Many literature sources investigate the relationship between WL and processing modes and WEDM quality indicators [6,8,19,20,35]. In particular, various options are proposed for the relation between the thickness of the modified layer and the roughness of the resulting surface, with the energy of discharge pulses, with the performance, and with the duration of the pulses. In this work, we estimated not the thickness of WL but the rate of formation of its volume V_w_. The data presented above in Table 3, Table 4 and Table 5 made it possible to determine the relation between V_w_ and the characteristics of the treatment process and the parameters of the AE signals. In Figure 7, the graphical representations show the relation of V_w_ to the WEDM performance and the set pulse duration.

The mathematical model of dependence V_w_ (S, T_on_) is represented by expression (3):V_w_ (S, T_on_)= −14.8344 + 0.9448·S + 1.4652·T_on_ + 0.0003·S^2^ − 0.0366·T_on_^2^ + 0.0028·S·T_on_(3)

For model (3), the SD value calculated from all experimental data is 3.9%. The graph in Figure 7c attracts attention to the fact that the relation between V_w_ and the performance of the process S is close to a directly proportional relation. At the same time, the influence of the T_on_ value is almost invisible. This is also seen in Figure 7b,d. Figure 8 shows the dependencies of the performance S on the A_1_ component and on T_on_, which can be compared with similar graphs for V_w_.

Comparing the graphs in Figure 8 with those in Figure 5c and Figure 7d, we can note an almost linear relation between the quality indicators S and V_w_ and the A_1_ component of the AE signal and a weakly expressed relation to the pulse width of the discharge current T_on_. It can also be noted that components A_2_ and A_3_ of the AE signal do not have a significant impact on performance compared to the A_1_ component. This effect on S is similar to that of AE components on V_w_, as shown in Figure 5d and Figure 6d.

### 3.2. The Relation between Surface Roughness, Performance, and AE Parameters

The roughness of the surface obtained after WEDM is also one of the indicators of the quality of processing. In the literature, the resulting value of Rz, as well as the thickness of WL, is associated with the energy of discharge pulses. In [19], it is stated that Rz has a linear relation with the thickness of the modified layer. You can find references to other types of dependencies, where the Rz indicator monotonically increases with increasing performance or pulse energy, but according to a power dependence, the exponent is 0.3–0.42 [20,27]. Therefore, the relation of Rz with performance, pulse parameters, and AE signals is interesting. Figure 9 shows the dependence Rz (S, T_on_).

The mathematical model of the dependence R_z_ (S, T_on_) is represented by the expression (4). The standard deviation error of the resulting model was 3.3%.
R_z_ (S, T_on_) = 33.1521 + 1.3312·S + 1.5394·T_on_ − 0.0083·S^2^ − 0.0432·T_on_^2^ − 0.0021·S·T_on_. (4)

If we compare the graphs for R_z_ in Figure 9 with similar graphs in Figure 7 for V_w_, it can be seen that there is no linear dependence of R_z_ on S. This dependence is extreme, but due to the sluggish nature of the extremum, it somewhat resembles saturation. The effect of T_on_ on R_z_ (Figure 9c) is more noticeable than the impact on S but remains relatively small within the CP area. In the literature, there is usually a monotonous increase in roughness with an increase in the performance of the WEDM process. Therefore, other relations of R_z_—for example, with the parameters of AE signals—are of interest. Figure 10 presents graphs showing the connection of the component signals A_1_ and A_3_ with R_z_.

The mathematical model with which the images in Figure 10 were built is represented by Expression (5):R_z_ (A_1_, A_3_) = 45.3701 + 2.0096·A_1_ − 1.6999·A_3_ − 0.0164·A_1_^2^ + 0.0048·A_3_^2^ + 0.0147·A_1_·A_3_
(5)

The standard deviation error of the model (5) was 6.9%. Figure 10b–d show that with small values of the A_1_ component, the influence of the A_3_ component is much more noticeable compared to that in Figure 6, where similar constructions were made for the model V_w_ (A_1_, A_3_). The relation of the high-frequency component with roughness, manifested in a decrease in R_z_ with an increase in amplitude A_3_ relative to the constant value of amplitude A_1_, can be explained by the rise in the power density of thermal energy in these modes. This causes an increase in the proportion of the evaporated substance of the workpiece compared to the melted substance, manifested in the growth of the AE signal at a high frequency [2,5]. However, a decrease in R_z_ is observed in moderate modes, where the amplitudes of A_1_ are small, which corresponds to a relatively low performance and growth rate of WL.

### 3.3. The Relation between WL and the Formation of Microcracks

While studying the modified layer, the issue of crack formation was raised. It is usually noted that WL is the source of crack formation in the base material of the product [6,8]. However, the study of pores and cracks in the white layer during the WEDM of a heat-resistant alloy showed a slightly different crack formation mechanism associated with the grain boundaries of the base material. Figure 11 shows photographs of thin sections taken with an optical microscope. The pictures clearly show the boundaries between the grains that outcrop the contact surface of the base material with WL. In these places, you can see the features in the white layer as pores and lines resembling cracks. However, there were no examples showing that the cracks in the WL continued in the grains of the base material.

It can be assumed that a crack in the white layer is formed due to the processes of heating and cooling the joint of materials with different values of the thermal expansion coefficient. At the same time, it is not a crack in the WL that generates a crack in the base material, but the movement of the boundaries between the grains forms a crack in the WL. The formation of pores in the WL near the outcrop of grain boundaries can occur due to an increased evaporation of the material in these places. Under the influence of thermal stresses, the pores can stretch and merge, forming centers of microcrack development. Figure 12 shows photographs taken with a scanning electron microscope (SEM) Tescan VEGA 3 LM (Tuscan Brno s.r.o., Brno, Czech Republic), indirectly confirming this hypothesis. Figure 12a shows the distribution of pores by WL. It can be noted that near the outcrop of the grain boundaries to the surface of the WL, pores are visible, represented in the photographs as dark spots. Under the thermal deformations of the base material and WL, the grain boundaries can move apart, stretching the WL material, which has a lower coefficient of thermal expansion. This leads to the deformation of the pores and the formation of cracks between them. Figure 12b demonstrates this situation. The center of the photograph shows a cascade of pores near the outcrop of the grain boundary to WL. A crack connecting them formed between the two pores closest to the boundary. If the deformation was somewhat more significant, one more pore could join these two pores, increasing the size of the resulting crack.

Thus, the above example shows that a crack can form not from the white layer towards the base material but from the boundary between the grains of the base material towards the WL. This example does not exclude other WL-related crack formation mechanisms but suggests that such a mechanism requires further study.

## 4. Conclusions

Experimental studies of the relation between the quality indicators of the surface layer of a workpiece made of a heat-resistant alloy and the characteristics of acoustic emission signals accompanying WEDM in different processing modes were carried out. As indicators of the quality of processing, the performance of the WEDM process, the rate of formation of volumes of the white layer, and the roughness of the resulting surface were studied. RMS amplitudes were recorded as acoustic emission signal parameters in three frequency ranges covering frequencies from 4 to 33 kHz. The properties of the surface layer change significantly as a result of WEDM, and this affects the quality of the finished product. The surface layer combines several layers, of which the white layer is given the most attention. This is because before carrying out further operations after WEDM, it is necessary to decide whether the white layer needs to be removed.

The results of the research can be the following:-Acoustic emission signals accompanying WEDM can be reliably recorded using an accelerometer installed on the machine table at a great distance from the processing zone in the frequency range up to 40 kHz.-As a parameter characterizing the formation of a white layer, a value proportional to the volume of the white layer formed per time unit was determined. Experiments have shown a close-to-linear relation between the rate of formation of the white layer and the RMS amplitude of the AE signal in a relatively low-frequency range of 4–8 kHz. The correlation with the AE signal components at higher frequencies was noticeably smaller.-Experiments have shown that the rate of formation of the volumes of the white layer is almost proportional to the performance of the WEDM process, and it practically does not depend on the pulse duration.-The relation of performance with acoustic emission signals is close to linear in the range of 4–8 kHz and is hardly noticeable at higher frequencies. The linear relation is more stable for high and medium performance, while the random component increases at low performance.-Surface roughness increases in WEDM performance, but this relation can only be approximated by a linear relation at low to medium performance. At higher performance, the roughness values reach saturation and practically do not change. From the duration of the pulses, the roughness varies within 10–15%, increasing in the region of 20 μs and decreasing for shorter and longer pulses.-The increase in the amplitude of acoustic emission signals in the high-frequency range compared to the amplitudes at low frequencies is associated with a decrease in roughness. This is especially noticeable at low and medium process performance.-Based on the results of studying the relationship between the parameters of acoustic emission signals and the performance and quality indicators of WEDM processes, it can be concluded that they can serve as additional diagnostic parameters used for the automatic control of the treatment process. This conclusion takes into account the possibility of acoustic emission signal parameters indicating an increase in the concentration of erosion products in the electrode gap in order to inform about the danger of short circuits and wire electrode breaks noted in other publications [2,5].-During the study of the white layer using a scanning electron microscope, an example of the development of cracks was found not from the white layer towards the base material but in the opposite direction. This allowed us to assume the existence of a mechanism for forming cracks in the white layer due to different coefficients of thermal expansion of the base material and the white layer. This leads to the stretching and consolidation of the pores of the white layer and the formation of cracks.

## Figures and Tables

**Figure 1 sensors-23-08288-f001:**
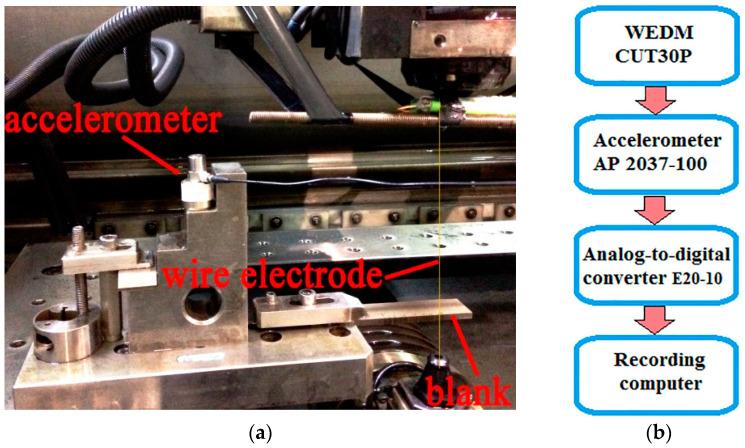
Registration of AE signals on the CUT 30P machine: (**a**)—processing area of the machine with an installed accelerometer; (**b**)—connection diagram of recording equipment.

**Figure 2 sensors-23-08288-f002:**
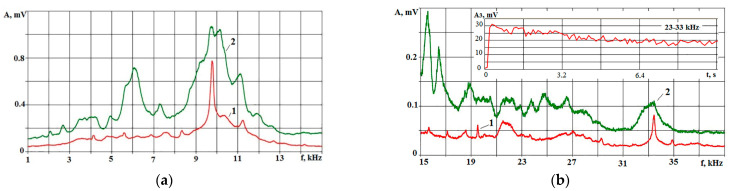
Spectra of AE signals in modes with minimum performance (1) and with maximum performance (2) in frequency ranges up to 15 kHz (**a**) and in the range of 15–40 kHz (**b**). The inset of figure (**b**) shows an example of recording an RMS signal amplitude in the range of 23–33 kHz over a period of 10 s.

**Figure 3 sensors-23-08288-f003:**
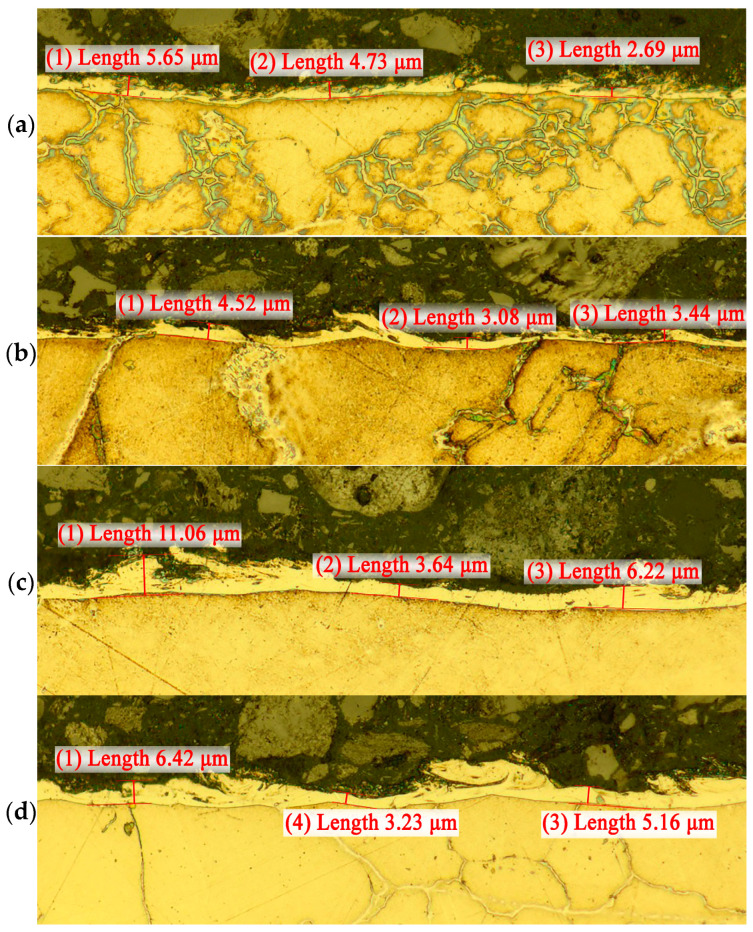
Examples of photographs of the white layer (WL) under different WEDM modes: (**a**)—experiment 3; (**b**)—experiment 9; (**c**)—experiment 4; (**d**)—experiment 8. The WL thickness measurement marks in the photo are shown in red.

**Figure 4 sensors-23-08288-f004:**
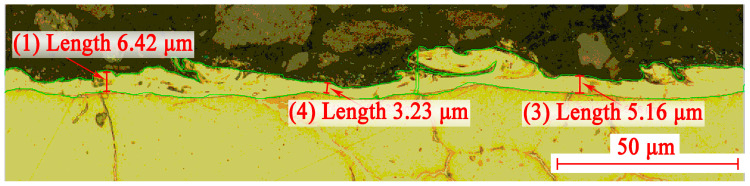
An example of highlighting the perimeter of the white layer (green line) in the photo of the section. Red numbers show the local thickness of the white layer.

**Figure 5 sensors-23-08288-f005:**
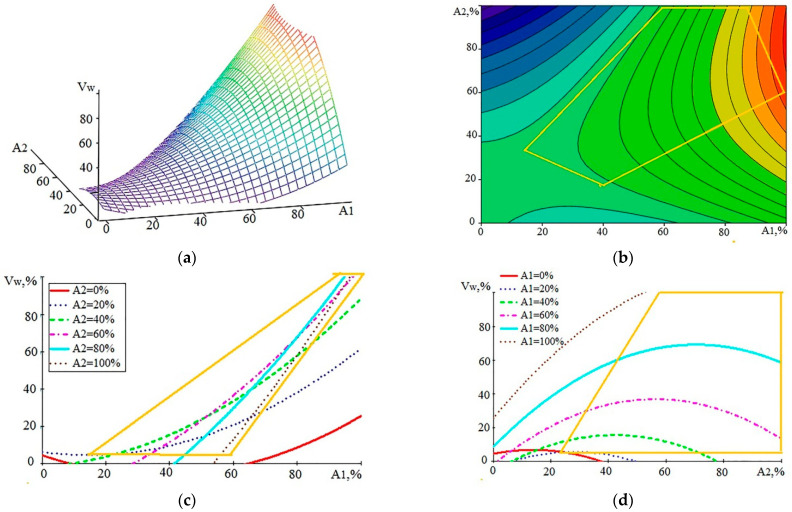
Graphical view of the dependence of V_w_ on the components of AE in the form of RMS values of the amplitudes A_1_ and A_2_: (**a**)—general view of the dependence; (**b**–**d**)—projections of lines of an equal level on coordinate planes (the yellow polygon is the perimeter of the region where the experimental points are located).

**Figure 6 sensors-23-08288-f006:**
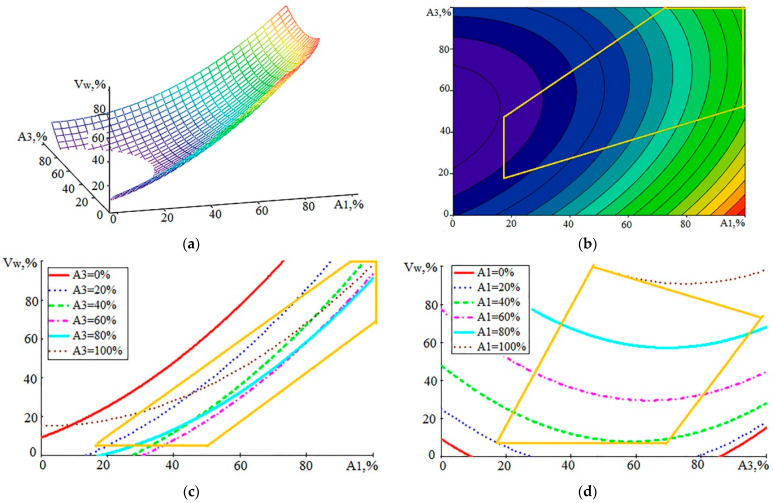
Graphical view of the dependence of V_w_ on the components of AE in the form of RMS values of the amplitudes A_1_ and A_3_: (**a**)—general view of the dependence; (**b**–**d**)—projections of lines of an equal level on coordinate planes (yellow polygon—CP area).

**Figure 7 sensors-23-08288-f007:**
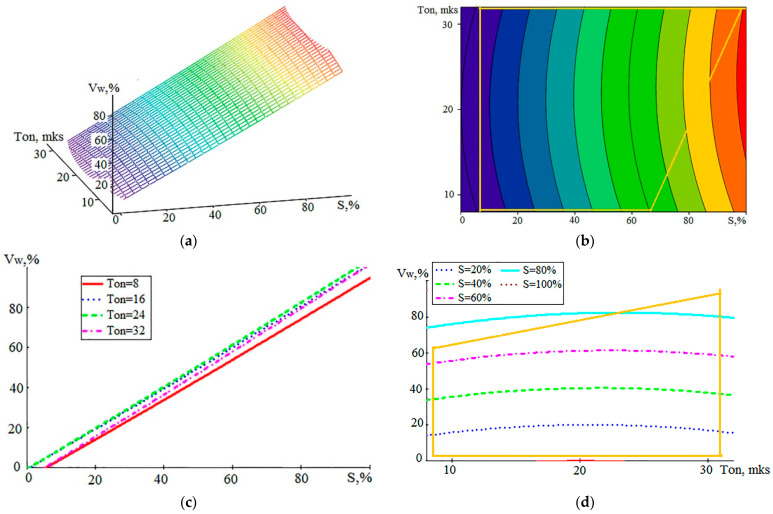
Graphical view of the dependence of V_w_ on the performance S and the set pulse duration T_on_: (**a**)—general view of the dependence; (**b**–**d**)—projections of lines of an equal level on coordinate planes (yellow polygon—CP area).

**Figure 8 sensors-23-08288-f008:**
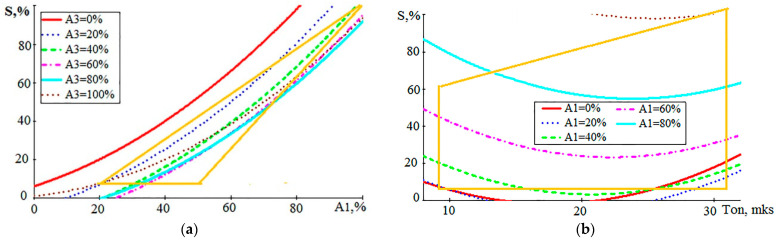
Graphical representation of the relation of the performance S with the amplitude A_1_ (**a**) and the duration of the discharge pulses T_on_ (**b**).

**Figure 9 sensors-23-08288-f009:**
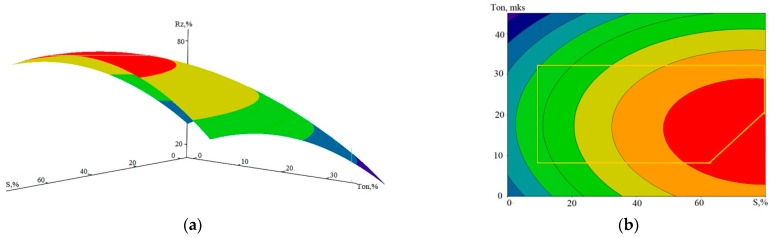
Graphical representation of the relation of the roughness R_z_ with the performance S and the duration of the discharge pulses T_on_: (**a**)—a general view of the relation; (**b**–**d**)—projections of lines of an equal level on coordinate planes.

**Figure 10 sensors-23-08288-f010:**
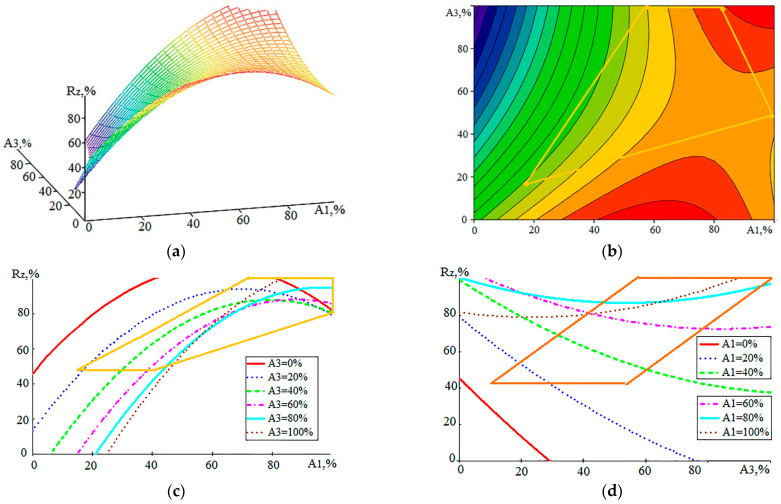
Graphical representation of the relation of the roughness R_z_ with the components of AE: (**a**)—a general view of the dependence; (**b**–**d**)—projections of lines of an equal level on coordinate planes.

**Figure 11 sensors-23-08288-f011:**
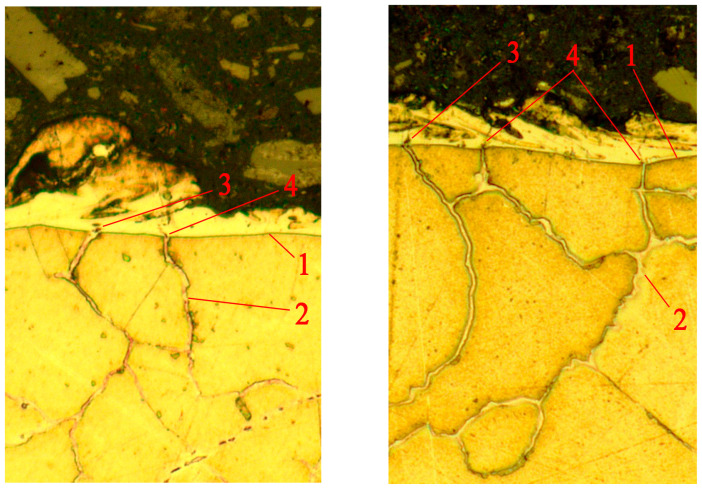
Photographs of the white layer with features in the vicinity of the outcrop of the grain boundaries to the contact surface with WL: 1—boundary of the white layer; 2—grain boundary; 3—pores in the vicinity of the grain boundary; 4—microcrack in the white layer (image width 50 µm).

**Figure 12 sensors-23-08288-f012:**
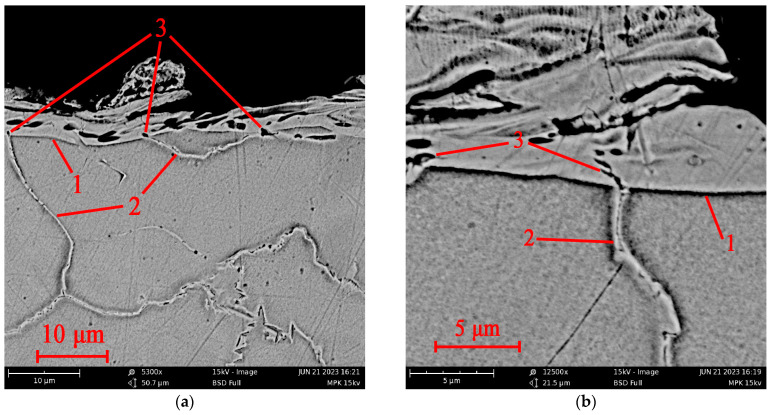
SEM photographs of pores at the WL junction with grain boundaries with low (**a**) and high (**b**) resolutions; 1—boundary of the white layer; 2—grain boundary; 3—pores in the vicinity of the grain boundary.

**Table 1 sensors-23-08288-t001:** Chemical composition of NiCr20TiA alloy.

**Chemical element**	Fe	C	Si	Mn	Ni	S	P	Cr	Ce	Ti	Al	B
**Mass fraction, %**	up to 4	0.07	0.6	0.4	69–78	0.007	0.015	19–22	0.02	2.4–2.8	0.6–1	0.01

**Table 2 sensors-23-08288-t002:** Accelerometer characteristics.

Accelerometer AP2037-100			
*K* _N_	Conversion coefficient	10	mV/m/s^2^
*f*	Linear frequency range	0.5–15,000	Hz
*f* _R_	Axial resonant frequency (more than)	45	kHz
Δ	Noise level, RMS [1 Hz–10 kHz] (less than)	0.0035	m/s^2^

**Table 3 sensors-23-08288-t003:** Main technical characteristics of the analog-to-digital converter (ADC) E20-10 (L’Card, Moscow, Russia).

Parameter	Value
ADC capacity	14 bits
Maximum ADC conversion frequency	10 MHz
Voltage measurement subranges	±3.0 V; ±1.0 V; ±0.3 V (independently adjustable for each channel)
Low-frequency conversion path bandwidth	0 Hz
Typical ADC channel signal-to-noise ratio	73 dB

**Table 4 sensors-23-08288-t004:** Main technical characteristics of the current sensor LA 100-P (LEM S.A., Meyrin, Switzerland).

Current Sensor LA 100-P			
*I* _PN_	Primary nominal RMS current	100	A
*I* _P_	Primary current, measuring range	0–±150	A
*I* _SN_	Secondary nominal RMS current	50	mA
*K* _N_	Turns ratio	1:2000	
*V* _C_	Supply voltage (±5%)	±12–15	V
*f*	Frequency bandwidth (−1 dB)	0–200	kHz
*ε*	Error *I*_PN_, TA = 25 °C (±12—15 V (±5%))	±0.70	%
*ε* _L_	Nonlinearity	<0.15	%

**Table 5 sensors-23-08288-t005:** Processing modes and WEDM performance.

Experiment No.	1	2	3	4	5	6	7	8	9	10	11	12	13	14	15	16
I, A	11	14	17	20	11	14	17	20	11	14	17	20	11	14	17	20
T_on_, mks	8	8	8	8	16	16	16	16	24	24	24	24	32	32	32	32
S, %	7.8	17.1	34.5	64.1	7.7	17.5	34.7	64.1	7.8	17.8	39.2	70.1	9.7	25.3	51.5	100

**Table 6 sensors-23-08288-t006:** RMS amplitudes of AE for three frequency ranges.

No.	1	2	3	4	5	6	7	8	9	10	11	12	13	14	15	16
**A_1_, %**	18	36.7	50.2	63.7	28.8	57.3	68.8	81.3	33.5	64	68.6	84.3	35.1	50.5	65.3	100
**A_2,_ %**	30	25	40	55	30	61	84	100	57	88	98	99	68	76	72	56
**A_3_, %**	19	35	48	59	28	54	70	79	34	64	90	100	45	66	80	46

**Table 7 sensors-23-08288-t007:** Data for calculating the rate of WL volume formation.

No.	Cutting Depth in 30 s in mm	WL Area at a Length of 197.4 µm in Pixels	WL Area Formed per Second in Pixels (Vw)	Vw, %
1	0.24	514,493	20,850.8	0.24
2	0.52	177,291	15,567.6	0.52
3	1.0	317,656	53,640	1.0
4	1.8	484,097	147,142	1.8
5	0.24	331,300	13,426.5	0.24
6	0.52	543,000	47,680	0.52
7	1.0	416,120	70,267	1.0
8	1.8	477,328	145,084.5	1.8
9	0.24	233,315	9455.5	0.24
10	0.52	498,234	43,749	0.52
11	1.09	583,232	107,349	1.09
12	1.9	523,730	168,032	1.9
13	0.3	282,365	14,304	0.3
14	0.74	312,023	38,990	0.74
15	1.5	477,715	121,002	1.5
16	2.88	483,983	235,372	2.88

**Table 8 sensors-23-08288-t008:** Results of measurements of real-time pulses (µs) and current values (A).

**Number of the experiment**	1	2	3	4	5	6	7	8	9	10	11	12	13	14	15	16
**Current, EDM**	11	14	17	20	11	14	17	20	11	14	17	20	11	14	17	20
Current *	20	40	50	60	20	40	60	60	20	40	60	60	35	40	60	60
**T_on_, EDM**	8	8	8	8	16	16	16	16	24	24	24	24	32	32	32	32
T_on_ *	6	6	7	6	8	8	8	8	21	21	21	21	133	163	162	136

* Results of measurements from additional sensors.

## Data Availability

Data sharing is not applicable to this article.

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
