# Peer review of "Display of WEDM Quality Indicators of Heat-Resistant Alloy Processing in Acoustic Emission Parameters"

_sensors, 2023, doi:10.3390/s23198288_

Round 1

Reviewer 1 Report

Reviewer report

1.       Methodology can be more understandably written. Such as, the procedure of the experiment can be systematically illustrated. For example, clarifications for specimen preparing procedure, AE measuring technique etc. can be added more clearly. The information regarding the experiment, data acquisition, data analysis is not clearly explained.

2.       Data acquisition results in different Experiments (1~16) are different. Why these differences have been come are unknown. I think the conditions of each experiment should be added clearly for understanding the results.

3.       Fractal dimensions analysis is commonly used for understanding the crack volume with AE parameters. How does the author(s) present the Novelty of the present paper?

4.       Author(s) should clarify more clearly about the applications of the mathematical models (1~5) in analyzing the results of the experiment. 

English should be imporved.

Author Response

Thank you for your comments and suggestions.

The following is a report on changes we have made in the revised manuscript to meet your suggestions

  1. Methodology can be more understandably written. Such as, the procedure of the experiment can be systematically illustrated. For example, clarifications for specimen preparing procedure, AE measuring technique etc. can be added more clearly. The information regarding the experiment, data acquisition, data analysis is not clearly explained.

Changes have been made to the article regarding the methodology for measuring AE signals. A diagram of the signal measurement channel and a table with the characteristics of the analog-to-digital converter have been added. Additional explanations are given regarding the method of preparing samples to obtain photographs of areas of the white layer. It was clarified that during the experiments the amplitude of the discharge current and the duration of the supplied pulses changed, other factors remained constant.

  1. Data acquisition results in different Experiments (1~16) are different. Why these differences have been come are unknown. I think the conditions of each experiment should be added clearly for understanding the results.

Table 4 (formerly Table 3) shows the values of the processing modes, which varied. It also shows the productivity values of the processing process, which were determined based on the results of measurements of the volumes of removed materials and processing time. The text clarifies that the amplitude of the discharge current and the duration of the supplied pulses changed.

  1. Fractal dimensions analysis is commonly used for understanding the crack volume with AE parameters. How does the author(s) present the Novelty of the present paper?

The article was not aimed at analyzing the mechanism of crack formation in the white layer, but when analyzing photographs using a scanning electron microscope, original results were obtained. In the literature there are descriptions of the mechanism of crack formation, where cracks initially form in the white layer and then move deeper into the surface. In the experiments conducted, another mechanism was noted when pores formed at grain boundaries serve as a source of cracks in the white layer under the influence of tensile stresses. This hypothesis is not related to acoustic emission, but it was decided to note it as a possible direction for further research.

  1. Author(s) should clarify more clearly about the applications of the mathematical models (1~5) in analyzing the results of the experiment. 

Clarifications have been made to the text, which indicate that with the help of mathematical processing (methodology for planning experiments), mathematical models were obtained, on the basis of which graphic images were constructed. The graphs made it possible to visually assess the nature of the obtained connections.

Thank you very much for your review.

Reviewer 2 Report

This paper aimed to study the relationship between the acoustic emission parameters and electrical discharge machining with the processing output characteristics. The topic of this manuscript is interesting. However, following shortages must be overcomed before publication.

1.      The literature review about the use of AE on mechanical engineering needs a great improvement as AE is an important measuring technique in the current research. There are many published papers that demonstrating the signal processing and the correlations between AE signals and material defects.

2.      It is necessary to show the details of the AE sensor and monitoring system used in the experiment. The operating frequency range of the AE sensor should also be provided.

3.      The sampling frequency and sampling length must be provided.

4.      In addition to the frequency domain waveform, I suggest the authors also provide the time-domain waveforms in Figure 2.

5.      The authors claimed thatThe purpose of this work was to study the interrelations of the acoustic emission parameters accompanying EDM.However, the authors only calculate the RMS and failed to extract enough parameters from AE signals and study the relationship between the multiple parameters and EDM. There are many time domain AE parameters such as count, amplitude, energy, information entropy, and frequency domain AE parameters such as peak frequency and centroid frequency. I suggest the authors calculate more AE parameters and tie them the EDM characteristics. Regarding the use of multiple AE parameters, the reviewer recommends the authors to cite the following papers and analyze more features.

https://doi.org/10.1016/j.compositesb.2020.108039

https://doi.org/10.1016/j.ijfatigue.2022.106860

https://doi.org/10.1016/j.engfracmech.2020.107083

https://doi.org/10.1016/j.ijpvp.2023.104998

6.      Please carefully check the scale bar in Figure 4.

7.      The conclusions part is too lengthy, and must be shortened.

Minor editing of English language required

Author Response

Thank you for your comments and suggestions.

The following is a report on changes we have made in the revised manuscript to meet your suggestions.

  1. The literature review about the use of AE on mechanical engineering needs a great improvement as AE is an important measuring technique in the current research. There are many published papers that demonstrating the signal processing and the correlations between AE signals and material defects.

The reviewer is absolutely right that AE parameters are widely used in mechanical engineering to monitor the condition of components and technological processes. The presented article is devoted to such a technological process as electrical discharge machining. Not much technical literature in English is devoted to acoustic emissions during electrical discharge machining. The authors have tried to indicate sources relevant to the topic of the article.

  1. It is necessary to show the details of the AE sensor and monitoring system used in the experiment. The operating frequency range of the AE sensor should also be provided.

The article was supplemented with a diagram of the channel for observing acoustic emission signals, technical characteristics of the sensor indicating the resonant frequency and linear range. The text indicates that signals can be monitored using the sensor used in the frequency range up to 79 kHz. The article also added a table with the characteristics of the analog-to-digital converter included in the AE observation channel.

  1. The sampling frequency and sampling length must be provided.

A clarification has been added to the article that the AE signals were recorded at a frequency of 1 MHz, and that the recording length was more than 30 seconds.

  1. In addition to the frequency domain waveform, I suggest the authors also provide the time-domain waveforms in Figure 2.

In Figure 2 there is an insert depicting the change in RMS signal amplitude in the range of 23-33 kHz for a period of 10 seconds from the start of cutting in the modes of experiment No. 16.

  1. The authors claimed that “The purpose of this work was to study the interrelations of the acoustic emission parameters accompanying EDM.”However, the authors only calculate the RMS and failed to extract enough parameters from AE signals and study the relationship between the multiple parameters and EDM. There are many time domain AE parameters such as count, amplitude, energy, information entropy, and frequency domain AE parameters such as peak frequency and centroid frequency. I suggest the authors calculate more AE parameters and tie them the EDM characteristics. Regarding the use of multiple AE parameters, the reviewer recommends the authors to cite the following papers and analyze more features.

The authors are grateful to the reviewer for pointing out sources where other AE parameters are analyzed. The authors have articles where the analysis of AE signals is carried out in the time domain; there are links to them in the References section. This paper analyzes the RMS of AE amplitudes in different frequency ranges. The square of the RMS amplitude is proportional to the signal energy that the reviewer mentions. Monitoring changes in RMS amplitude over time for different AE frequency ranges gave a good result for determining the increase in the concentration of erosion products in the electrode gap and preventing wire electrode breaks (Sergey N. Grigoriev, Mikhail P. Kozochkin, Marina A. Volosova, Anna A. Okunkova and Sergey V. Fedorov, Vibroacoustic Monitoring Features of Radiation-Beam Technologies by the Case Study of Laser, Electrical Discharge, and Electron-Beam Machining. Metals 2021, 11, 1117. https://doi.org/10.3390/met11071117). It is difficult to present in one article all the results of studies of WEDM processes using the analysis of various AE parameters due to the limitation of the volume of the article, but the authors will try to describe other results in future publications.

  1. Please carefully check the scale bar in Figure 4.

In Figure 4, the operator’s marks are shown in red when measuring the local thickness of the white layer (vertical line) on a microscope. In the figure below there are arrows on the corresponding labels.Figure 4.jpg

  1. The conclusions part is too lengthy, and must be shortened.

When drawing conclusions, the authors proceeded from the fact that not all readers will read the article in detail with graphs, formulas and other details. Look at the abstract and conclusions. Therefore, the conclusions were made more detailed. One page with conclusions is acceptable.

Thank you very much for your review.

Reviewer 3 Report

The paper presents an interesting study. A few corrections need to be made.

  1. 1. The "Experimental Setup and Procedure" section must be included to describe how the experiments were conducted.

  2.  
  3. 2. The sensors and measuring devices used in the experiment need to be listed, and their accuracy and precision should be tabulated.

  4.  
  5. 3. The novelty of the study needs to be explicitly stated at the end of the introduction.

Author Response

Thank you for your comments and suggestions.

The following is a report on changes we have made in the revised manuscript to meet your suggestions.

  1. The "Experimental Setup and Procedure" section must be included to describe how the experiments were conducted.

A diagram of the AE signal measurement channel has been added to the article, a table with the characteristics of the equipment has been added, clarifications have been added to the text based on your comments and the comments of other reviewers.

  1. The sensors and measuring devices used in the experiment need to be listed, and their accuracy and precision should be tabulated.

Additional tables have been introduced indicating the main characteristics of the equipment and accuracy.

  1. The novelty of the study needs to be explicitly stated at the end of the introduction.

At the end of the introduction, a paragraph is inserted reflecting the scientific novelty of the presented article.

Thank you very much for your review.

Round 2

Reviewer 2 Report

Thank you. I have no more comments.

I have no more comments.